# Improving Antimicrobial Properties of Biopolymer-Based Films in Food Packaging: Key Factors and Their Impact

**DOI:** 10.3390/ijms252312580

**Published:** 2024-11-22

**Authors:** Sonia Wardejn, Stanisław Wacławek, Gabriela Dudek

**Affiliations:** 1Department of Physical Chemistry and Technology of Polymers, Faculty of Chemistry, Silesian University of Technology, Strzody 9, 44-100 Gliwice, Poland; sw300312@student.polsl.pl; 2Institute for Nanomaterials, Advanced Technologies and Innovation, Technical University of Liberec, Studentska 1402/2, 461 17 Liberec, Czech Republic; stanislaw.waclawek@tul.cz

**Keywords:** biofilms, food packaging, antimicrobial, environmentally friendly

## Abstract

Biodegradable films derived from polysaccharides are increasingly considered eco-friendly alternatives to synthetic packaging in the food industry. The study’s purpose was to improve the antimicrobial properties of biopolymer-based films made from starch, chitosan, alginate, and their blends (starch/chitosan and starch/alginate) and to evaluate the effects of modifiers, i.e., plant extracts, plasticizers, cross-linking agents, and nanofillers. Films were prepared via the Solution Casting Method and modified with various plasticizers, calcium chloride, oxidized sucrose, and nanofiber cellulose (NC). Chestnut, nettle, grape, and graviola extracts were tested for antimicrobial activity against *Staphylococcus epidermidis*, *Escherichia coli*, and *Candida albicans*. The film’s mechanical and hydrophilic properties were studied as well. The chestnut extract showed the strongest antimicrobial properties, leading to its incorporation in all the films. The chitosan films displayed better antibacterial activity against Gram-positive than Gram-negative bacteria but were ineffective against *C. albicans*. NC significantly improved the mechanical and antimicrobial properties of the chitosan films. The alginate films, modified with various plasticizers cross-linked with calcium chloride, demonstrated the highest antimicrobial efficacy against *E. coli.* The starch films, cross-linked with oxidized sucrose, exhibited slightly lower antimicrobial resistance due to a more compact structure. Films such as ALG6 and ALG5, including plasticizers EPGOS and PGOS, respectively, indicated optimal hydrophilicity and mechanical properties and achieved the best antimicrobial performance against all the investigated microorganisms. All these findings highlight the potential of these biodegradable films for food packaging, offering enhanced antimicrobial activity that prolongs shelf life and reduces spoilage, making them promising candidates for sustainable food preservation.

## 1. Introduction

Growing environmental concerns over synthetic packaging have led to increased interest in biopolymer-based packaging materials [1]. There has been a notable trend of developing biodegradable food packaging materials using polysaccharides and other natural polymers. Derived from renewable resources such as plant biomass and animal waste products, natural polymers offer a sustainable alternative to fossil-based plastics. Natural polymers are inherently biodegradable, breaking down under atmospheric factors or microbial action into low-molecular-weight compounds that cause no harm to the environment. In contrast, synthetic polymers like plastics degrade into microplastics, which may accumulate in ecosystems and enter food chains, posing potential risks to wildlife and human health. These qualities position natural polymers as a preferred choice for developing safe, eco-friendly antimicrobial films for food packaging materials.

Furthermore, the rise in pathogenic bacteria in food packaging has heightened concerns over food safety, as contamination during packaging, transport, and storage can lead to foodborne illnesses, spoilage, and economic losses. Pathogens like *E. coli*, *S. aureus*, *Salmonella*, *L. monocytogenes*, and *C. albicans* are often linked to outbreaks due to inadequate packaging that fails to prevent microbial growth [2,3]. In response, biopolymer-based films with natural antimicrobial agents offer a promising solution by actively inhibiting microbial contamination, thereby enhancing food safety and shelf life. Traditional packaging materials, while providing physical barriers, lack antimicrobial properties and may contribute to contamination by degrading into microplastics that support bacterial biofilm formation [4].

Polysaccharides are considered the leading candidates for replacing oil-based polymers in food packaging. These biopolymers are quite advantageous materials—they are easy to form, biodegradable, safe, cheap, accessible, environmentally friendly, and act as a matrix for incorporation additives with specific functionalities for packaging films, i.e., antimicrobial properties, low extraction cost, and good film-forming properties. Renewable polysaccharides include cellulose, nanocellulose, hemicellulose, chitosan, starch, pectin, alginate, and others [5]

Polysaccharides have been developed into films and coating for the packaging of different food products such as meat, fish, fruit, and vegetables [6]. One of the primary limitations of polysaccharides is that they provide an energy source for living organisms, so bacteria can also consume them. Nevertheless, chestnut extract can mitigate the impact of bacteria consuming polysaccharides. Consequently, such packaging films have the potential to extend the shelf life of food products. In order to prevent microbial growth and oxidation, different types of additives can be added to polysaccharide-based films and coatings for food packaging, e.g., extracts, plasticizers, and nanoparticles.

Among the various plant extracts, *Vitis vinifera* leaf extract and *Hibiscus sabdariffa L*. extract have become the most commonly used additives in biofilms due to their demonstrated effectiveness in enhancing antimicrobial properties [4,5]. Capar [7] added *Vitis vinifera* leaf extract to alginate-based edible films, leading to improved structural integrity, antimicrobial activity, and antioxidant properties of these films. Hernández-Hernández et al. [8] incorporated *Hibiscus sabdariffa* L. extract into starch-based films, significantly enhancing the material’s antimicrobial activity. As a result of a high level of polyphenolic compounds in such extract, the adhesion to the bacteria cell wall was compromised, promoting the cytoplasmic membrane disruption and releasing the cell’s internal components. Additionally, polyphenols extracted from many types of tea are increasingly popular as additives in biodegradable films, especially chitosan films. These polyphenols contribute to enhancing the antimicrobial characteristics of these films [9].

In recent years, silver nanoparticles and zinc oxide nanoparticles have appeared as key components in biofilm research [7,8,10]. Silver nanoparticles are widely recognized for their potent antimicrobial properties [11]. Ponnusamy et al. [12] integrated silver nanoparticles (AgNPs) into chitosan-based films combined with an ethanolic extract of Gracilaria crassa (GCE). This integration enhanced the physical and mechanical properties of the films through the synergistic effects of chitosan, GCE, and AgNPs, resulting in film wraps that significantly extended the shelf life of shrimp nuggets stored under refrigeration. Alqarni et al. [13] incorporated co-doped zinc oxide nanoparticles (ZnO NPs) into chitosan and gelatin nanocomposites, significantly improving their antimicrobial activity. This enhancement is attributed to the production of reactive oxygen species (ROS) during photocatalysis under UV or visible light, which oxidizes intracellular proteins, membranes, and DNA in bacterial cells. The antimicrobial activity of the co-doped zinc oxide nanoparticles is further increased by their ability to generate free hydroxyl radicals. Furthermore, Mirres et al. [14] combined green synthesized zinc oxide nanoparticles with babassu coconut extracts, creating a synergistic effect that improved the antimicrobial properties of the material. This combination effectively reduced the microbial count of *S. aureus* in cooked turkey ham samples stored under refrigeration for 7 days.

In terms of plasticizers, there is a growing focus on using deep eutectic solvents (DESs) as innovative plasticizers. DES are advantageous due to their non-toxic nature and environmentally friendly properties. They provide excellent biocompatibility, biodegradability, and thermal stability. Additionally, specific DESs are easy to produce and can dissolve polysaccharides effectively, providing superior plasticizing properties compared to traditional plasticizers [15]. For instance, Yu et al. [16] incorporated DES-tomato extracts into chitosan films, resulting in improved ultraviolet shielding and enhanced antioxidant activity. This led to better storage quality and ultimately extended the shelf life of strawberries. Similarly, Jiang et al. [17] introduced a thymol-based deep eutectic solvent as a plasticizer in chitosan films. This modification resulted in enhanced antibacterial and antioxidant activities, as well as improved hydrophobic properties. Consequently, the DES-plasticized chitosan films were highly effective in prolonging the shelf life of grapes.

In this study, we characterize the antimicrobial properties of biopolymer-based films for food packaging made of various biopolymer matrices from different natural sources such as starch, chitosan, and alginate. All these biopolymers were selected to investigate the impact of various biopolymer matrices on antimicrobial properties. Originating from diverse sources, they are widely studied in the literature, allowing us to benchmark our biopolymer-based films against existing research.

To characterize antimicrobial activity properly, we also take into account other factors that could affect the performance of the investigated films. According to previous studies [18,19,20,21,22], achieving antimicrobial effectiveness in films requires adding specific modifiers into the film matrix, such as nanofillers, cross-linking agents, plasticizers as well as plant extracts. These modifiers are widely available and well-described in the literature for their effectiveness in enhancing antimicrobial activity. We selected these additives to compare published findings with our experimental results. Films’ antimicrobial properties were examined against the most common Gram-positive and Gram-negative bacteria, as well as yeast. The specific impact of each additive is discussed. Furthermore, mechanical and hydrophilic properties were determined in the case of antimicrobial activity.

## 2. Results

The aim of this study was to develop and optimize antimicrobial biopolymer-based films using polysaccharides—starch, chitosan, alginate, and their blends—for sustainable food packaging applications. The novelty of this research lies in its systematic evaluation of various modifiers, including plant extracts, plasticizers, cross-linking agents, and nanofillers, as well as their combined impact on enhancing the antimicrobial, mechanical, and hydrophilic properties of these films. The findings from this study demonstrate that these modified biopolymer-based films exhibit promising antimicrobial efficacy, positioning them as viable candidates for future eco-friendly food packaging solutions.

### 2.1. Impact of Plant Extracts on Antimicrobial Properties

Figure 1 shows the representative photograph displaying the inhibition zone assay results against *S. epidermidis*, *E. coli*, and *C. albicans.*

Among all the plant extracts tested, the chestnut extract exhibited the strongest antimicrobial properties against both Gram-positive (*S. epidermidis* ATCC12228) and Gram-negative bacteria (*E. coli* ATCC25922), as well as yeast (*C. albicans* ATCC18804) [23].

Notably, the chestnut extract indicates a slightly stronger inhibitory effect on Gram-positive bacteria. Similarly, Wang et al. [24] also examined the antibacterial activity of chestnut shell polyphenols against *B. subtilis*, *P. fragi*, and *E. coli*, among which *B. subtilis* was the most susceptible. In their study, the inhibitory effect on Gram-positive bacteria was stronger than on Gram-negative bacteria. Furthermore, chestnut shell polyphenols damage the bacterial cell wall and membrane, compromising structural integrity and increasing membrane permeability. As a result, this leads to cell content leakage, the inhibition of TCA cycle gene expression, reduced enzyme activity, hindered respiration and energy metabolism, and decreased ATP synthesis, ultimately inhibiting cell proliferation and causing bacterial cell death. In another study, Štumpf et al. [25] examined the antimicrobial properties of chestnut extract, particularly focusing on its tannin content. Here, tannins were isolated from the chestnut extract, and their effects on *E. coli* were evaluated. The results indicated that the chestnut tannins effectively inhibited bacterial growth by extending the lag phase and decreasing the bacterial growth rate. Consequently, chestnut extract can be used in food preservation. The other plant extracts, i.e., nettle, grape, and graviola showed significant antibacterial activity against *S. epidermidis*, though their efficacy was weaker compared to the chestnut extract. For instance, the chestnut extract has stronger antimicrobial efficacy than nettle [26], grape [21], and graviola [27] extracts owing to its high tannin content, particularly ellagitannins [28], which effectively disrupts microbial cells by binding to proteins, interfering with enzymes, and damaging cell membranes [29]. These tannins also deprive bacteria of essential nutrients, providing broad-spectrum activity against both Gram-positive and Gram-negative bacteria and reducing the risk of resistance.

All the extracts more effectively inhibited the growth of *S. epidermidis* compared to *E. coli*, indicating a higher antibacterial efficacy against Gram-positive strains. The difference in antimicrobial activity between Gram-positive and Gram-negative bacteria may be primarily attributed to the structural differences in their cell walls [30,31]. The bacterial cell wall is a rigid structure that covers the cytoplasmic membrane and gives structural support to bacteria. Specifically, Gram-positive bacteria have thick cell walls, consisting of several layers of peptidoglycan and surface glycopolymers such as teichoic acids. In contrast, Gram-negative bacteria are additionally protected by an outer membrane composed of lipopolysaccharide that restricts the penetration of antimicrobial agents. Thus, the thick but unprotected layer of Gram-positive bacteria lacks an outer membrane, making them easily accessible to antimicrobial compounds, i.e., polyphenols, flavonoids, and alkaloids found in plant extracts, which target the cell wall of the bacteria [27,31,32].

In the case of the remaining plant extracts, Garofulić et al. [33] isolated nettle (*Urtica dioica*) leaf polyphenols by using pressurized liquid extraction (PLE), analyzed their individual phenolic profile by UPLC MS2, and for their antioxidant capacity by ORAC assay. PLE extract showed antimicrobial activity against *P. fragi*, *C. jejuni*, *S. aureus*, and *Shewanella* strains. As a result, nettle leaf extract could be used as a potent agent for prolonging the shelf life of foods and reducing foodborne infections. Similarly, Sterniša et al. [34] prepared an ethanolic extract of the nettle and performed a phytochemical analysis to identify its bioactive components. Six main compounds were identified, including caffeoylmalic acid and five flavonoid glycosides. The nettle ethanolic extract demonstrated in vitro antimicrobial activity against specific strains of *Pseudomonas* and *Shewanella* in broth cultures and fish meat homogenate. Moreover, Flórez et al. [31] reinforced the nettle extract by the high UV barrier, which was incorporated into the chitosan matrix, resulting in enhanced antioxidant activity in the films. Oliveira et al. [32] proved the antimicrobial activity of grape (*Vitis vinifera*) pomace extract against *S. aureus*, *B. cereus*, *E. coli*, and *P. aeruginosa*. Due to the best properties of the chestnut extract, it was chosen and examined as an antimicrobial plant extract for all the investigated films.

### 2.2. Impact of Other Modifiers on Antimicrobial Properties of Biopolymers Films

In our study, the same amount of chestnut extract was added to all the films, allowing us to attribute the difference in antibacterial properties to the biopolymer matrix and the presence of modifiers.

The antimicrobial properties of the investigated films, summarized in Figure 2, are shown against representative model microorganisms, specifically targeting Gram-positive bacteria (*S. epidermidis*) in Figure 2a, Gram-negative bacteria (*E. coli*) in Figure 2b, and yeast (*C. albicans*) in Figure 2c.

The films, i.e., chitosan, starch, alginate, and their blends, i.e., starch/chitosan and starch/alginate, were investigated. The modifiers having an impact on antimicrobial properties were incorporated:(a)Nanofiber cellulose, and glycerol as a plasticizer for CH.(b)Oxidized sucrose as a cross-linking agent, and glycerol as a plasticizer for ST.(c)Calcium chloride as a cross-linking agent, and various plasticizers, i.e., glycerol; epoxidized soybean oil; epoxidized palm oil; mixed esters of propylene glycol and acetic acid; mixed esters of propylene glycol, oleic acid, and succinic acid; epoxidized mixed esters of propylene glycol; and oleic acid and succinic acid for ALG.(d)Oxidized sucrose as a cross-linking agent, and glycerol as a plasticizer for blends, i.e., STCH and STALG.

As the control samples (CTRL CH, CTRL ST, and CTRL ALG), pure films without chestnut extract and other modifiers were examined.

The films’ characteristics are summarized in Table 1 including their polymer matrices, added plasticizers, and control samples.

#### 2.2.1. Impact of Polymer Matrix on Antimicrobial Activity

As can be seen in Figure 2, the antimicrobial properties of the investigated films composed of biopolymers, i.e., chitosan, starch, or alginate, are diversified and strongly influenced by the characteristics of the biopolymer matrix.

The chitosan films, i.e., CH and CHNC, show stronger antimicrobial resistance towards *S. epidermidis* compared to the *E. coli* study, possibly owing to the differences in the bacterial cell walls (Figure 2a,b). The antifungal effect was observed for all the investigated films (Figure 2c) apart from the chitosan films, although chitosan itself is known to exhibit antifungal properties. Chitosan is a weak base, insoluble in water and organic solvents at neutral pH, which restricts its intrinsic antifungal activity and limits the release of the chestnut extract in the biopolymer matrix [35]. The antimicrobial properties of chitosan against *S. epidermidis*, *E. coli*, and *C. albicans* are attributed to its cationic structure. The positive charge of chitosan is due to the protonation of its amino groups (-NH_2_ to -NH_3_^+^) under acidic conditions, which enhances its solubility by enabling it to form soluble salts with acids. This cationic property is key to its interaction with microbial cell membranes, disrupting their integrity and leading to antimicrobial activity.

Hosseinnejad et al. [36] have confirmed this mechanism, providing clear evidence that the positive charge of chitosan disrupts microbial membranes and impairs cellular functions, leading to its potent antimicrobial activity. Several mechanisms contribute to the antimicrobial activity of chitosan. First, the positively charged amine groups in chitosan interact with the negatively charged bacterial cell wall, increasing membrane permeability and causing the leakage of intracellular contents, leading to cell death. Second, chitosan’s chelating properties enable it to selectively bind with metals (e.g., K⁺, Ca^2^⁺) and negatively charged molecules such as lipopolysaccharides (LPSs) in Gram-negative bacteria, as well as anionic biomolecules such as nucleic acids and proteins. This disrupts ionic balance, increases transmembrane potential, and inhibits metabolic enzymes by blocking their active sites, which interferes with replication and transcription [36,37]. Lastly, chitosan is able to enter the nucleus, bind to DNA, and block RNA synthesis, further inhibiting microbial growth [38].

In our study, Gram-positive bacteria are highly sensitive (Figure 2a), probably because of the structural differences in the cell walls between Gram-positive and Gram-negative bacteria. This phenomenon may explain the stronger antibacterial properties of chitosan-based films against Gram-positive than Gram-negative bacteria [39].

The Gram-positive bacterial cell wall is characterized by a thick peptidoglycan layer (20–80 nm), making up to 90% of its dry weight. This layer, along with negatively charged teichoic and lipoteichoic acids, creates an acidic microenvironment by interacting with cations. This local acidity supports the solubility of chitosan, which dissolves well in environments with pH values below 6.5 [40].

We also observed that the addition of nanofiber cellulose (NC) to the chitosan matrix significantly enhances its antimicrobial properties against Gram-positive and Gram-negative bacteria as well as yeast. The CHNC composite combines the high mechanical strength of cellulose nanofibers with the inherent antimicrobial nature of chitosan, resulting in a synergistic effect. Further details on the influence of nanofillers are provided in Section 2.2.4.

The alginate films demonstrated the strongest antimicrobial activity against *S. epidermidis* and *E coli* compared to the chitosan films (Figure 2a,b). The protonated chitosan amino groups facilitate hydrogen bonding between adjacent polymer chains, leading to a more compact structure and stronger chain interactions [41]. This compact structure leads to decreased free volume between polymer chains, causing less effective migration of chestnut extract particles, resulting in worse antimicrobial properties compared to the alginate films.

Interestingly, the alginate films showed greater resistance to *E. coli* compared to *S. epidermidis*, indicating stronger antibacterial properties against the Gram-negative bacteria in this study. As shown in Figure 2c, the antifungal properties of the alginate films are better than those of the chitosan films. Unmodified alginate films typically do not exhibit intrinsic antimicrobial properties because they do not possess any specific functional groups that can interact with bacterial cell membranes or inhibit microbial growth. The antimicrobial efficacy of alginate films is imparted by adding modifiers with antibacterial activity, i.e., metal nanoparticles, essential oils, or other bioactive compounds [42]. The hydrophilic nature of alginate film facilitates the migration of additives, such as chestnut extract, to its surface. In fact, alginate swells in a water environment, which increases free volume between polymer chains, and as a result, the migration of chestnut extract is expanded [43]. Upon reaching the surface, the chestnut extract becomes active, resulting in inhibiting bacterial proliferation. Tannins present in chestnuts contribute significantly to this process by extending the bacterial lag phase and reducing the overall growth rate of bacterial populations. Motelica et al. [44] incorporated varying concentrations of ZnO nanoparticles loaded with citronella essential oil (CEO) into alginate films to enhance antibacterial properties. The study showed a synergistic antibacterial effect of ZnO and CEO against both Gram-negative (*E. coli*, *S. Typhi*) and Gram-positive bacteria (*B. cereus*, *S. aureus*), with the highest efficacy observed against *B. cereus*. These films demonstrated broad antibacterial activity and hold promise for food preservation applications, particularly in preserving cheese.

Furthermore, the antimicrobial efficacy is affected not only by the polymer matrix composition and additives, like plant extracts, but also by the plasticizers integrated into the film. These plasticizers increase the free volume between polymer chains, facilitating the migration and sustained activity of chestnut extract particles. We incorporated six different plasticizers into the alginate films, resulting in varied levels of antimicrobial activity. More information about these plasticizers is provided in Section 2.2.2.

We also observed that physical cross-linking, achieved by soaking alginate films in calcium chloride, enhances film swelling and loosens the polymer structure, thereby facilitating the migration of antimicrobial agents like chestnut extract. Alginate films cross-linked with calcium chloride inhibit bacterial activity by interfering with bacterial physiological functions and preventing biofilm formation. Further details about these cross-linking agents are available in Section 2.2.3.

The starch films, such as ST and STALG, exhibit stronger antimicrobial resistance against *E. coli* compared to *S. epidermidis* (Figure 2a,b). Specifically, the antimicrobial effectiveness of the starch films is greater than the activity of the chitosan films against *E. coli*, while the chitosan films demonstrate better performance against *S. epidermidis*. However, compared to the alginate films, the starch films are less effective against *E. coli*. ST performs similarly to ALG in combating *S. epidermidis*, with both achieving a bacterial log reduction of 3.8. Furthermore, starch-based blends such as STCH and STALG show enhanced antimicrobial properties against *S. epidermidis*. In terms of antifungal activity, pristine starch outperforms certain alginate films (e.g., ALG, ALG3, and ALG4) in inhibiting *C. albicans*.

In comparison to the ST film, STCH shows greater antimicrobial effectiveness against *S. epidermidis*, highlighting the impact of incorporating chitosan into the starch polymer matrix. Moreover, ST and STALG exhibit stronger antifungal activity against *C. albicans* than STCH (Figure 2c). Additionally, this study showed that the starch-based films were more resistant to *E. coli* bacterial pathogens than the chitosan-based films. Starch films in their native state are characterized by a semi-crystalline granular structure, tending to be brittle and highly sensitive to moisture owing to their hydrophilic nature. Pure starch does not have an inherent ability to inhibit microbial growth. To impart antimicrobial properties to starch-based films, other polymers as well as additional agents such as silver nanoparticles, essential oils, plant extracts, and cross-linking agents are incorporated into the biopolymer matrix. We added oxidized sucrose as a cross-linking agent, promoting antimicrobial activity through acetal formation within the film. This process induces the leakage of cellular components, ultimately resulting in bacterial cell death. We provide more information about the impact of cross-linking agents in Section 2.2.3.

Antimicrobial nanobiocomposites are gaining attention as a safe way to extend food shelf life by inhibiting microbial growth. Incorporating nanostructured ingredients into the biopolymer matrix is a promising approach to enhance these materials’ functional properties. Singh et al. [45] incorporated starch nanocrystals (SNCs) and lemongrass oil nanoemulsion (LNE) into corn-starch nanocomposite films [46]. The results indicate that SNC-/LNE-loaded films exhibit higher antimicrobial activity against *E. coli* and *S. aureus* than SNC films.

As observed in Figure 2b, combining different polymer matrices and preparing blend films such as STCH is beneficial in the case of enhanced antibacterial resistance to Gram-positive bacteria (*S. epidermidis*). Chitosan, similarly to starch, is a non-toxic, biodegradable, biocompatible polysaccharide with film-forming properties. Its high solubility in acidic conditions enhances antimicrobial activity and strengthens binding to Gram-positive bacterial cell walls. Alves et al. [47] incorporated chitosan into starch film, combining the thermoplastic properties of starch with the antioxidant and antimicrobial activities of chitosan. As a result, the blend’s antioxidant activity is significantly higher than the pristine chitosan, reaching an inhibition ranging from 71% to 79% due to the increased mobility between polysaccharide chains and turning the films less dense.

When considering all the investigated films, it was observed that the cationic polymers are more effective on Gram-positive bacteria than the anionic groups. Compared to STCH, STALG exhibits slightly improved antimicrobial resistance towards *E. coli*. Consequently, the prepared films have more inhibitory effects against Gram-negative bacteria compared to Gram-positive bacteria. Additionally, alginate tends to have more spaced flexible chains than chitosan, contributing to efficient chestnut extract migration to the surface of the film. Blending starch with sodium alginate helps overcome some of the starch’s limitations, such as its brittleness and high water sensitivity. Alginate also contributes to better water retention and plasticity, making the films more durable in humid environments. Moreover, these blends often exhibit enhanced biocompatibility and can be modified further for active packaging with antimicrobial and antioxidant properties through the incorporation of bioactive additives like essential oils or nanoparticles [48].

Zhang et al. [49] prepared an antibacterial composite film by blending starch, sodium alginate, and montmorillonite, fortified with star anise oil as the bacteriostat. As a result, the incorporation of star anise oil gave antimicrobial activity, resisting *E. coli*. Using this composite as packaging film could reduce the decay rate of fresh cherry tomatoes.

In our study, we observed that incorporating modifiers into the film matrices significantly enhanced their antimicrobial activity. Detailed information on these modifiers, i.e., plasticizers, cross-linking agents, and nanofillers, is provided in the following Section 2.2.2, Section 2.2.3 and Section 2.2.4.

#### 2.2.2. Impact of Plasticizers on Antimicrobial Activity

As can be observed in Figure 2. the films with the most effective antimicrobial activity are alginate films, i.e., ALG6, ALG5, and ALG2, including plasticizers, i.e., EPGOS (epoxidized mixed esters of propylene glycol, oleic acid, and succinic acid), PGOS (mixed esters of propylene glycol, oleic acid, and succinic acid), and ESO (epoxidized soybean oil), respectively. This phenomenon is related to their ability to increase the free volume between polymer chains, which enables chestnut extract particles to migrate and stay active.

The alginate films, especially ALG6 and ALG5, displayed superior antimicrobial activity against *E. coli* with a bacterial log reduction of 4.7. In addition, they showed strong effectiveness against *S. epidermidis* and *C. albicans* (Figure 2a,b). These films contain bioplasticizers, namely EPGOS and PGOS, respectively, which indicates a synergistic effect between the added plasticizer and chestnut extract. Furthermore, ALG2 exhibited stronger antimicrobial activity compared to ALG3, which contains epoxidized palm oil as a plasticizer (EPO). ESO contains a higher amount of linoleic acid compared to EPO used in ALG3.

According to Yoon et al. [50], linoleic acid exhibits the most potent activity to inhibit the growth of Gram-positive bacteria, including *S. aureus*, by damaging its cell membrane. This fatty acid can cause the lysis of *S. faecalis* as well. Moreover, linoleic acid induces the cell lysis of *H. pylori* belonging to Gram-negative bacteria. ALG4 demonstrated good antimicrobial properties containing as a plasticizer the mixed esters of propylene glycol and acetic acid. Among all the investigated films with plasticizers, ALG containing glycerol has the worst antimicrobial activity. Kester et al. [51] found that glycerol-containing films absorb more water, leading to increased moisture content and diminished film antimicrobial effectiveness. This contrasts with other plasticizers that might create less hydrophilic films. Moreover, the addition of glycerol to alginate films can increase flexibility and decrease density, potentially weakening the film’s ability to form a strong barrier against bacteria. More porous or softer films can be more susceptible to bacterial colonization, reducing their antimicrobial activity. Hong et al. [52] discussed how the choice of plasticizer can affect the barrier and mechanical properties of films, thereby influencing their antimicrobial functionality. Films with more flexible and less dense structures, often resulting from glycerol, tend to exhibit reduced antimicrobial properties.

Glycerol is also used as a plasticizer in chitosan and starch films. Muscat et al. [53] studied glycerol’s film-forming behavior in starch-based films, finding that higher glycerol concentrations increased the moisture content and water permeability. Specifically, at 40% glycerol content, the tensile strength significantly decreased due to reduced intra-molecular bonding, favoring the formation of hydrogen bonds with starch molecules instead of facilitating bonding between the polymer chains. Consequently, this phenomenon allowed for greater flexibility but also weakened the tensile strength. Compared to the alginate film containing glycerol, the starch film has similar antimicrobial activity against *S. epidermidis*, while the chitosan film is slightly worse (Figure 2a). For *E. coli* (Figure 2b) the alginate film exhibits much stronger antibacterial activity in comparison to starch and chitosan film, which is the least effective. While comparing blends, STCH demonstrates stronger antimicrobial activity against *S. epidermidis*, whereas STALG shows better antimicrobial properties against *E. coli.* When it comes to *C. albicans*, the starch films have stronger antifungal activity than the alginate films. The chitosan films do not exhibit antifungal properties.

The incorporation of suitable plasticizers is crucial for enhancing the mechanical properties of materials and vapor barrier performance, which subsequently impacts their antimicrobial effectiveness. Plasticizers are commonly added to enhance flexibility, especially in packaging applications. However, concerns about the migration of plasticizers into food, particularly if the plasticizers have toxic effects, have led to an increasing preference for bioplasticizers. Bouftou et al. [54] compared widely used plasticizers, i.e., glycerol, tris (2-ethylhexyl) phosphate, and rosemary essential oil, added to cellulose acetate films for food packaging. Their findings revealed that rosemary oil had the lowest migration rate into food stimulants. Additionally, rosemary essential oil plays dual roles as a plasticizer, which enhances mechanical properties, as well as an antimicrobial agent, showing efficacy against *E. coli* and *S. aureus*. Yaman et al. [55] developed antimicrobial PLA composite films incorporating epoxidized soybean oil (20%), spruce resin (15%), ZnO (0.1%), and thyme/clove essential oils (5% and 10%, respectively). They found that this combination enhanced the films’ antimicrobial efficacy against *E. coli* and *S. aureus*. Spruce resin acted as both a plasticizer and antimicrobial agent, ZnO as an antimicrobial nanofiller, and thyme oil contributed additional antimicrobial effects.

As can be seen, the antimicrobial properties of the film are influenced not only by the plasticizer used but also by the composition of the polymer matrix from which the film is made and the presence of cross-linking agents.

#### 2.2.3. Impact of Cross-Linking Agents on Antimicrobial Activity

The following modifiers incorporated into films were cross-linking agents, i.e., calcium chloride and oxidized sucrose in the alginate and starch films, respectively. They were added not only to improve antimicrobial properties but also to enhance their mechanical and barrier properties.

As can be seen, the impact of the cross-linking agents on the antimicrobial properties of the films is significant, with the differences observed between physical and chemical cross-linking. The alginate films cross-linked with calcium chloride (physical cross-linking) inhibit bacterial activity by disrupting bacterial physiological functions and preventing biofilm formation [56]. In contrast, the starch films cross-linked with oxidized sucrose (chemical cross-linking) undergo acetal formation, leading to the leakage of cellular components and bacterial cell death [57]. While the chitosan films were not cross-linked, the combination of starch and chitosan exhibited enhanced antibacterial activity against *S. epidermidis* and *E. coli*, suggesting a synergistic effect of incorporating chitosan into the starch matrix, as well as the positive influence of the cross-linking agent. Physical cross-linking proves more effective for antimicrobial activity than chemical cross-linking, as it allows for greater film swelling and the loosening of the polymer structure and as a result, facilitates the migration of antimicrobial agents such as chestnut extract.

In contrast, chemical cross-linking creates a more compact structure, restricting the mobility of antimicrobial agents between polymer chains and reducing the overall effectiveness. Zhang et al. [58] showed that tannic acid effectively cross-links polysaccharide films, i.e., starch, cellulose, chitosan, and alginate, improving their mechanical strength and antimicrobial activity. The antibacterial effect of tannins is due to phenolic hydroxyl groups, which act as bacteriostatic agents for most bacteria. Mechanisms include the following: (i) inhibiting bacterial iron uptake, (ii) disrupting cell wall integrity, and (iii) preventing biofilm formation [59]. Comparably, Khan et al. [60] incorporated genipin as a cross-linking agent to nanocomposite films, resulting in the inhibited growth of psychrotrophs, mesophiles, and *Lactobacillus* spp. in fresh pork loin meats, and increased the microbiological shelf-life of meat sample by more than 5 weeks. Moreover, the combination with gamma irradiation acted in synergy with the film.

#### 2.2.4. Impact of Nanofillers on Antimicrobial Properties

As can be seen in Figure 2, a significant modifier incorporated into the chitosan films is nanofiber cellulose (NC), resulting in the composite material CHNC, which shows enhanced antimicrobial properties against *S. epidermidis*, *E. coli*, and *C. albicans*. For CHNC, the observed bacterial log reductions were 0.3, 1.2, and 0.8 CFU/mL, respectively.

According to Mithilesh Yadav et al. [61], adding nanofillers to a polymer matrix significantly improves mechanical properties such as tensile strength, flexibility, and durability. Enhanced mechanical strength results in stronger antimicrobial properties, which is shown in Figure 2. The composite structure of CHNC leverages the high mechanical strength of cellulose nanofibers and the antimicrobial, biocompatible nature of chitosan. Additionally, the interaction between cellulose and chitosan leads to better dispersion of nanofibers within the chitosan matrix, reinforcing the film’s structure and reducing brittleness. The incorporation of nanofiber cellulose also contributes to better film stability and moisture resistance, making it a highly efficient composite for advanced applications. Studies show that chitosan nanofillers enhance antimicrobial properties, reducing bacterial motility and impacting the ability of pathogens to form biofilms, which are crucial for their survival and virulence. Zou et al. [62] enhanced chitosan films’ properties by adding lignin nanoparticles (LNPs) and acylated soy protein isolate nanogel (ASPNG), improving tensile strength, barrier properties, UV blocking, and antibacterial effects. The hydrophilic ASPNG swells in water, modifying the film structure and enabling better LNP release.

### 2.3. Impact of Mechanical Properties on Antimicrobial Properties

The mechanical properties, including tensile strength (TS) and elongation at break (EB), as illustrated in Figure 3, show significant correlations with the antimicrobial activity of the films. Films with higher tensile strength values (Figure 3a), such as CH (8 MPa), CHNC (9 MPa), and STCH (10 MPa), generally exhibit stronger antimicrobial effectiveness against *S. epidermidis*. This suggests that a more rigid structure may disrupt the Gram-positive bacteria cell wall more effectively than Gram-negative bacteria. Biratu et al. [63] observed that in pectin films, tensile strength improves as the concentration of pectin solution increases from 1% to 4% because of the formation of a stronger polysaccharide network. Interestingly, films with higher tensile strength—especially those including additives like propolis, honey, and glycerol—were more effective at suppressing Gram-positive bacteria (such as *S. aureus* and *L. monocytogenes*) than Gram-negative bacteria (like *E. coli* and *P. aeruginosa*). The enhanced antimicrobial activity of the films against Gram-positive bacteria is supposedly due to their simpler peptidoglycan-based cell walls, which are easily disrupted by more rigid films. In contrast, Gram-negative bacteria have a more complex outer membrane, making them more resilient to the films due to their additional protective barrier [64]. Amjad Ali et al. [65] incorporated pomegranate peel (PGP) into starch-based films, significantly enhancing their mechanical properties, such as tensile strength. Notably, improvements in tensile strength were directly linked to stronger antimicrobial activity, with a more pronounced effect against Gram-positive bacteria (*S. aureus*) compared to Gram-negative bacteria (*Salmonella*). As PGP concentration increased, the films’ ability to inhibit bacterial growth also improved, particularly against *S. aureus*. The increased strength and rigidity of the film enhance its antibacterial efficacy, particularly against Gram-positive bacteria, by allowing better contact and sustained pressure on their thick cell walls, leading to greater mechanical disruption. In contrast, the complex outer membrane of Gram-negative bacteria like *Salmonella* makes them less vulnerable to mechanical stress. Additionally, the blend STCH was cross-linked by oxidized sucrose, while nanofiber cellulose was incorporated into CHNC, both contributing to a higher TS value as well as a compact biopolymer matrix. Chitosan nanocomposite films, i.e., CHNC, demonstrated enhanced antimicrobial activity (9 MPa, 57% EB) against both *S. epidermidis* and *E. coli* when compared to pristine chitosan (8 MPa, 50%EB). This suggests that even slight improvements in tensile strength and elongation at break can positively impact antibacterial effectiveness. In contrast, higher EB values (Figure 3b), as seen in ST (78% EB) and STALG (90% EB) compared to CH (50% EB), CHNC (57% EB), and STCH (25% EB), correlate with a better performance against *E. coli*, indicating that greater film flexibility may facilitate interactions with Gram-negative outer membrane. Interestingly, films with low TS and EB, such as ALG6 (1.5 MPa, 8% EB), demonstrated the strongest overall antimicrobial activity, particularly against *S. epidermidis*, *E. coli*, and *C. albicans*. This suggests that physical cross-linking, which alters the flexibility of alginate films, can be advantageous. Loosening the polymer structure may enhance the release of antimicrobial agents, such as chestnut extract, allowing them to reach the film surface and remain active. The alginate films such as ALG (2.4 MPa, 15% EB) also performed well against *E. coli*, showing that moderate TS value with very low EB correlates with higher effectiveness against *E. coli*. This phenomenon indicates that stronger, less flexible films may have enhanced antibacterial activity against this strain. Aloui et al. [66] significantly improved the mechanical properties of sodium alginate (NaAlg) films by adding gallnut extract (GE) as a cross-linking agent in varying NaAlg/GE weight ratios. Increasing the GE content from 0 to 25 wt% enhanced the tensile strength and elongation at break by 48–103% and 135–185%, respectively, without compromising thermal stability. The GE addition also boosted the films’ antioxidant capacity and exhibited strong antibacterial activity against both Gram-positive and Gram-negative bacteria, as well as antifungal properties against *S. aureus*, *E. coli*, *A. niger*, and *P. digitatum*. These findings indicate that the film’s enhanced mechanical properties, achieved through the incorporation of gallnut extract, directly contribute to its antimicrobial effectiveness. Abutalib et al. [67] investigated the enhancement of the mechanical and antibacterial properties of chitosan (Cs) and polyethylene oxide (PEO) films by incorporating silver (Ag) and titanium dioxide (TiO_2_) nanoparticles. The addition of these doped mixed nanoparticles to the polymer blend led to significant improvements in mechanical properties, such as tensile strength and elongation at break. Moreover, the inclusion of Ag/TiO_2_ nanoparticles increased the antibacterial efficacy of the Cs/PEO blend. Specifically, at a nanoparticle concentration of 0.3% Ag and 0.8% TiO_2_, the antibacterial activity index against *E. coli*, *S. aureus*, *C. albicans*, and *A. niger* was observed to be 32%, 45.8%, 77.8%, and 92%, respectively.

### 2.4. Impact of Hydrophilic Properties on Antimicrobial Properties

The water interaction characteristics, including swelling degree (SD) and contact angle, as illustrated in Figure 4, play a critical role in determining antimicrobial film activity. As shown in Figure 4a, films with the lowest SD values—CH, CHNC, and ST (40%, 50%, and 60%, respectively)—do not exhibit the strongest antimicrobial activity against *S. epidermidis*, *E. coli*, and *C. albicans*. In contrast, films with the highest SD values, such as STALG and STCH, demonstrate significantly better antimicrobial performance against all the investigated microorganisms. It is worth noting that the samples based on two polysaccharides, i.e., STCH and STALG, showed much higher SD values than these polymers separately. Mathew et al. [68] investigated the swelling properties of starch/chitosan blend films, observing that the swelling degree increased with higher starch content. This is due to both the hydrophilic nature of starch and the natural swelling behavior of chitosan in acetic acid. This phenomenon is linked to the enhanced protonation of the amino groups in chitosan, leading to the formation of -NH_3_^+^ groups at an acidic pH of two, which promotes film swelling. Furthermore, the combination of starch and chitosan results in films with a greater swelling capacity than the films made from pure starch or chitosan alone. ALG consisting of glycerol as a plasticizer shows the highest SD value among the investigated alginate films, but it also exhibits the weakest antimicrobial activity against the same microbial strains. This phenomenon indicates that excessively high SD can lead to over-swelling, potentially degrading the film’s mechanical integrity and reducing its ability to maintain consistent contact with the target surface, such as food packaging. However, the most effective antimicrobial films, namely ALG6, ALG5, and ALG2, display moderate SD values. This suggests that films with moderate swelling degree values are optimal, as they allow sufficient water absorption to promote the diffusion of antimicrobial agents without compromising the film’s structural integrity. A slightly increased SD can enhance the diffusion of antimicrobial agents, such as chestnut extract, improving bioactivity by facilitating their migration to the film’s surface. Achieving the right balance between swelling capacity and cross-linking density is crucial for optimizing both the structural stability and antimicrobial function of these films. Ayse Su Giz et al. [69] investigated the effect of glycerol and cross-linking with calcium chloride on the swelling degree of alginate films. This study shows that swelling in water decreased from 66% for (10 Gly-0.5 Ca) to 51% for (10 Gly-2 Ca) as the amount of calcium chloride increased. Simultaneously, the swelling degree was slightly increased from 66% to 68% by increasing the glycerol content in the alginate films.

The water contact angle (Figure 4b) plays a key role in influencing the antimicrobial properties of films by determining their hydrophilicity or hydrophobicity, which affects the release of antimicrobial agents, consequently enhancing the films’ antimicrobial effectiveness. A contact angle (θ) below 90° denotes a hydrophilic surface, while an angle value above 90° indicates hydrophobicity [70]. Among the tested films, only CHNC exhibited a contact angle of approximately 100°, indicating its hydrophobic nature. The addition of nanofillers to pristine chitosan significantly increased the contact angle from 70° to 100°, making CHNC more hydrophobic. The presence of nanocellulose fiber reduces water absorption, limits swelling, and enhances the structural integrity of the film. However, the increased hydrophobicity may also hinder the diffusion of water-soluble antimicrobial agents, such as chestnut extract, by limiting their release from the film matrix. Consequently, this can reduce interaction with microbial cells and, thereby, antimicrobial effectiveness. Despite this, CHNC still demonstrated stronger antimicrobial activity against *S. epidermidis*, *E. coli*, and *C. albicans* compared to standard chitosan (CH) films, likely due to the enhanced structural integrity of the polymer matrix. Lakovaara et al. [71] studied the impact of deep eutectic solvents (DESs) as a reaction medium for the modification of nanocellulose and all-cellulose composite (ACC) films. By utilizing DES for the esterification of cellulose nanofiber (CNF) and ACC films with n-octylsuccinic anhydride (OSA), they significantly increased the hydrophobicity of both films. The modified CNF-DES/OSA and ACC-DES/OSA films exhibited contact angles of 51° and 60°, respectively, compared to the lower contact angles of the unmodified CNF and ACC films, which were 37° and 36°. The films with the lowest contact angles, STCH and ST (approximately 45° and 55°, respectively), displayed the most hydrophilic characteristics. These highly hydrophilic surfaces facilitated greater water absorption and swelling, enhancing the diffusion of antimicrobial agents to the film surface and improving antimicrobial performance against *E. coli* and *C. albicans* compared to the chitosan films. However, excessively low contact angles can lead to over-swelling, compromising the film’s structural integrity and limiting its contact with microorganisms. This phenomenon may explain the reduced effectiveness of the ST films against *S. epidermidis*. All the alginate-based films displayed moderate contact angles, which fall between 50° and 90°. The highest contact angles were observed in ALG2 and ALG3 (82° and 79°, respectively), which could be attributed to the structure of alginate, and the presence of plasticizers such as epoxidized oils (ESO and EPO, respectively). These plasticizers likely contribute to the strong antimicrobial activity against *S. epidermidis*, *E. coli*, and *C. albicans*. Nevertheless, the films with the highest overall antimicrobial efficacy, ALG6 and ALG5, demonstrated lower contact angles (67° and 54°, respectively), indicating that a moderate contact angle enhances antimicrobial effectiveness. This balance allows the film to absorb water without over-swelling, thus preventing structural degradation. This allows the efficient release of antimicrobial agents such as chestnut extract in moist environments, which significantly improves films’ antimicrobial properties.

### 2.5. Biodegradation of Biopolymer-Based Films

The study on the soil degradation of the prepared films revealed that all the films fully decomposed within 15 days. Although we used three types of biopolymer matrix, the biodegradability was similar. This is likely because all the matrices examined are polysaccharides, giving them similar characteristics. Moreover, the investigated modifiers come from natural sources, thus they did not worsen the degradation rate.

The degradation of the investigated films was also evaluated by immersing them in PBS at 37 °C. The results, presented in Figure 5, demonstrate how the type of biopolymer matrix influences the degradation behavior of these films. The film with the highest biodegradability was alginate, which decomposed in less than 12 h. The other films did not degrade completely within 72 h, exhibiting different weight loss rates. The most stable film was chitosan, with a weight loss of 23.5 ± 1.5% after 72 h. In contrast, the starch film exhibited a weight loss three times greater than chitosan, amounting to 69.2 ± 1.8%. Blending matrices combines the properties of each biopolymer. For the starch/chitosan blend, the weight loss was intermediate between the individual polymers, measuring 32.8 ± 1.6%. A similar trend was observed for the starch/alginate blend, which exhibited a weight loss of 50.5 ± 1.7%. Comparable results for biopolymer degradation have been previously reported in [72,73,74,75].

## 3. Materials and Methods

### 3.1. Materials

Starch was acquired from Heuschen & Schrouff OFT B.V. (Landgraaf, The Netherlands). Chitosan (30–100 cps, MW = 250,000, DD ≥ 90%) was sourced from Sigma-Aldrich (St. Louis, MI, USA). Sodium alginate (Brookfield viscosity 350–550 mPas, c = 1 wt.% at 20 °C) was obtained from Acros Organics (Branchburg, NJ, USA). Acetic acid (99.5–99.9%) was purchased from POCH S. A. (Gliwice, Poland). Chestnut extract Farmatan (≥76% tannins) was supplied by Tanin Sevnica (Sevnica, Slovenia). Calcium chloride (purity ≥ 96%) was provided by Avantor Performance Materials (Radnor, PA, USA). Glycerol was manufactured by Nortchem (Los Angeles, CA, USA). Oxidized sucrose was prepared using sucrose from Pfeifer & Langen Polska (Poznań, Poland), sodium periodate (>98%) from Acros Organics, and barium chloride from STANLAB Sp. J. Lublin (Lublin, Poland). Epoxidized soybean oil and epoxidized palm oil were acquired from Inbra Indústrias Químicas LTDA (Sao Paulo, Brazil) and Malaysian Palm Oil Board (Kajang, Malaysia), respectively. Propylene glycol, cyclohexene, and toluene (all pure p.a.), along with formic acid (85.0%), hydrogen peroxide (30.0%), disodium hydrogen phosphate dihydrate, and sodium hydrogen carbonate (pure p.a.), were purchased from Chempur (Piekary Śląskie, Poland). Oleic acid (90.0%) was supplied by Alfa Aesar (Ward Hill, MA, USA), and succinic acid (≥99.5%) by POL-AURA (Zabrze, Poland). Methanesulfonic acid (>99.0%) was obtained from TCI (Zwijndrecht, Belgium). Nanofibrillated cellulose (10–20 nm wide, 2–3 µm length) was purchased from Nanografi Nano Technology (Ankara, Türkiye).

### 3.2. Preparation of the Antimicrobial Films

In order to study the antimicrobial properties, biopolymer films were prepared using the Solution Casting Method. Three polymer matrices with additives were considered: starch, chitosan, and alginate. The polysaccharide of our choice was dissolved in proper conditions. The solution (46 g) was homogenized and then poured on the smooth surface, i.e., waxed Petri plates (12 × 12 cm each plate). The obtained films ca. 50 μm were dried (24 h) at room temperature and peeled off (Figure 6).

#### 3.2.1. Sodium Alginate Films

Aqueous sodium alginate solution (1%, *w*/*w*) was prepared by dissolving sodium alginate in water. After obtaining a clear solution, a chestnut extract (0.75%, *w*/*v*) and a selected plasticizer* (30%, *w*/*w* based on the mass of alginate) were added. The mixture was stirred overnight on a magnetic stirrer (IKA, Staufen, Germany) at 1000 rpm and room temperature (24 °C). Afterwards, the mixture was homogenized at 6000 rpm for 5 min with an homogenizer (Ultra-Turrax T50, IKA, Staufen, Germany) and left overnight. After homogenization, the film solutions were kept in the fridge overnight at 5 °C. The mixture was poured onto waxed Petri dishes and left at room temperature (24 °C) for drying. Then, alginate films were cross-linked by 40 mL 2.5% calcium chloride solution for 2 h. The film was rinsed with distilled water and laid out on a paper towel to prevent the film from wrinkling. The abbreviations of the alginate films are presented in Table 2.

#### 3.2.2. Plasticizer Mixtures

Plasticizer mixtures were synthesized according to [23,57] through esterification and epoxidation reactions. The esterification processes, involving propylene glycol with acetic acid, propylene glycol with oleic acid, and succinic acid, as well as the epoxidation of mixed esters derived from propylene glycol, oleic acid, and succinic acid, were conducted in a 500 or 1000 cm³ glass reactor. The reactor was equipped with a mechanical stirrer, temperature controller, Dean–Stark trap for the esterification reactions, reflux condenser, and a dropping funnel for specific reactions.

#### 3.2.3. Starch Films

To prepare starch-based films, a 5% (*w*/*w*) starch solution was made by first dissolving starch powder in water heated to 80 °C. Once the solution became transparent, the chestnut extract was added at a concentration of 0.75% (*w*/*v*), along with glycerol (used as a plasticizer) at 40% (*w*/*w*) relative to the starch content. This blend was then stirred continuously overnight at 1000 rpm on a magnetic stirrer (IKA, Staufen, Germany) at ambient temperature (24 °C). Later, the solution was subsequently homogenized at 6000 rpm for 5 min with an Ultra-Turrax T50 homogenizer (IKA, Staufen, Germany) to ensure uniform dispersion. To induce a cross-linking reaction, oxidized sucrose was introduced at 15% (*w*/*w*) of the starch mass, and the mixture was then heated to 90 °C and stirred for an additional 30 min. After mixing, the solution was stored overnight in a refrigerator at 5 °C to allow trapped air bubbles to escape. The prepared solution was then poured onto wax-coated Petri dishes and left to dry at room temperature (24 °C). Once the drying was complete, the starch films were carefully removed from the plates and placed in an air oven at 160 °C for 4 min to finalize the drying process.

##### Oxidized Sucrose

Oxidized sucrose was synthetized according to the procedure outlined by Wang et al. [76]. In short, 6.50 g of sucrose and 12.90 g of sodium periodate were dissolved in 200 mL of distilled water. The solution was continuously stirred at 1000 rpm at room temperature (24 °C) for 24 h using a magnetic stirrer (IKA, Staufen, Germany). After the reaction, approximately 7 g of barium chloride was incorporated into the solution, which was then stirred for an additional hour at 5 °C to promote the complete precipitation of byproducts. The mixture was subsequently filtered, and the clear filtrate was collected and stored at 5 °C for later use.

#### 3.2.4. Chitosan Films

Chitosan solution (2%, *w*/*w*) was prepared by dissolving chitosan in an acetic acid solution (1%, *v*/*v*). After obtaining a clear solution, various additives, i.e., the chestnut extract (0.75% *w*/*v*); plasticizers, i.e., glycerol (30%, *w*/*w* based on the mass of chitosan); mixed esters of propylene glycol and acetic acid (30%, *w*/*w* based on the mass of chitosan); and nanofiber cellulose were added. The mixture was stirred overnight on a magnetic stirrer (IKA, Staufen, Germany) at 1000 rpm and at room temperature (24 °C). Subsequently, the mixture was homogenized at 6000 rpm for 5 min with a homogenizer (Ultra-Turrax T50, IKA, Staufen, Germany) and left overnight. Afterward, the film solutions were kept in the fridge overnight at 5 °C. The mixture was cast on Petri dishes and left at room temperature (24 °C) for drying. In order to obtain biocomposite, the nanofibre cellulose (NC) was used as a filler, which may improve the properties of chitosan.

#### 3.2.5. Blend of Starch/Chitosan and Starch/Alginate Films

The procedure for preparing blends, i.e., the starch/chitosan and starch/alginate films, is the same as described for the individual polysaccharide films. Before adding the proper additives, starch with chitosan and starch with alginate were mixed. The abbreviations of the individual and blend films are presented in Table 3.

### 3.3. Measurement of Antimicrobial Properties

#### 3.3.1. Plant Extracts Antimicrobial Activity

Before preparing polysaccharide-based films, the properties of commercially available plant extracts were determined. First, the quick research of antimicrobial activity against model Gram-negative bacteria (*E. coli* ATCC25922), Gram-positive bacteria (*S. epidermidis* ATCC12228), and yeasts (*C. albicans* ATCC18804) of the plant extracts was performed. The plant extracts derived from chestnut, grape seeds, nettle, and graviola were tested using an inhibition zone assay. The control sample was double-distilled water.

#### 3.3.2. Polysaccharide Films’ Antimicrobial Activity

The antimicrobial activity of films (Figure 7) was determined against Gram-negative bacteria (*E. coli* ATCC25922), Gram-positive bacteria (*S. epidermidis* ATCC12228), and yeasts (*C. albicans* ATCC18804). These bacterial isolates, as well as yeast, were opted for in our study because of their usage in microbiology as model organisms. Both bacteria have the ability to create biofilms. Moreover, *E. coli* belongs to Psychrophiles [77], capable of growing and proliferating in cold temperatures such as refrigerated temperatures—typical for the fridge, where food may be stored. The Microdilution Assay was used to assess the antimicrobial activity of the polysaccharide films. The samples (10 mm in diameter) were placed in 12-well plates, which contained 200 μL M9 minimal medium supplemented with glucose as a sole carbon source. Then, 20 μL of the targeted bacterial culture was added to each well. Prior to placing bacterial cultures, all the targeted bacterial isolates were normalized to 10^4^ CFU/mL by serial dilution following co-culture with each film.

Briefly, a single bacterial colony was grown in 10 mL of Mueller–Hinton broth (MHB) at 37 °C overnight. After incubation, a 1 mL portion of each bacterial culture was collected by centrifugation at 7000× *g* for 3 min at 4 °C, then resuspended in a sterile saline solution. The bacterial density was adjusted to a 0.5 McFarland standard (approximately 1.5 × 10^8^ CFU/mL) using a densitometer. The normalized cultures were then incubated again overnight at 37 °C with shaking at 150 rpm. Following this, 20 µL of the bacterial suspension was added to each well in a 12-well plate, and the prepared films were analyzed in each well. After incubation, the bacterial cultures were serially diluted in distilled water and plated on LB agar to determine the CFU/mL of the target bacteria. Each experiment was conducted in triplicate, with the results expressed as the log10 of the mean CFU/mL.

A statistical analysis of significance was performed using an unpaired, two-tailed Student’s *t*-test (JMP software v.5; SAS Institute Inc., Cary, NC, USA). The CFU/mL values of the bacteria co-cultured with the films containing the chestnut extract were compared to the control films (alginate, starch, and chitosan films without extract). *p*-values below 0.05 were considered statistically significant. Additionally, a One-Way ANOVA was conducted to compare the responses of different model organisms—*S. epidermidis*, *E. coli*, and *C. albicans*—to the films’ samples. All the *p*-values were below 0.05, indicating significant differences among the tested film types (chitosan, starch, and alginate). These results suggest that *S. epidermidis*, *E. coli*, and *C. albicans* respond differently to the presence of chestnut extract in the films.

### 3.4. Measurement of Mechanical Properties

The mechanical properties of the films, specifically the tensile strength (TS) and elongation at break (EB), were evaluated using an Instron 4466 testing machine. The tests were conducted at a speed of 5 mm/min under room temperature conditions. For testing, the films were cut into strips measuring 80 mm in length and 20 mm in width. The final measurements for tensile strength and elongation at break were calculated by taking the average of ten individual measurements.

### 3.5. Measurement of Swelling Degree

The swelling degree (SD) of the films was measured using a gravimetric approach in three distinct steps. First, square samples with a 1 cm^2^ area were cut from the films and initially weighed on an analytical balance, recording this mass as *M*1. These samples were then placed in an oven at 100 °C for 24 h to dry completely, after which they were re-weighed to obtain *M*2. Following drying, the samples were submerged in 30 mL of distilled water and kept at room temperature (24 °C) for 24 h to allow for swelling. After this period, they were removed, and their weight was measured again, and recorded as *M*3. Finally, the samples were dried once more at 100 °C for an additional 24 h and weighed for a final mass *M*4. This entire process was repeated three times for each sample to ensure accuracy, and the average values were used for calculations. The swelling degree (SD) was calculated using the following equation:SD (%) = (*M*3 − *M*2) / *M*2 × 100(1)

### 3.6. Measurement of Contact Angle

To determine the contact angle of water on the film surfaces, an optical contact angle meter (OCA15, DataPhysic, Filderstadt, Germany) with contour analysis capabilities was used at room temperature (24 °C). A small drop of deionized water (1 µL) was precisely placed onto each film surface. The contact angle values were obtained by taking the mean of ten independent measurements.

### 3.7. Measurement of Biodegradation

The biodegradability of the films was tested using a soil degradation method. Small dry sample pieces (2 × 3 cm) were buried 5 cm below the soil surface in plastic containers. To maintain soil moisture at 40%, about 10 mL of water was added to the containers each day. After incubating for 15 days at 25 ± 1.0 °C, the samples’ final weight was measured. The rate of degradation in the soil was determined by calculating the percentage of mass lost compared to the initial weight of the samples.

We also conducted the hydrolytic degradation of the investigated films; samples with a surface area of 1 cm^2^—composed of chitosan, starch, and alginate, as well as their blends—starch with chitosan or alginate—underwent a systematic evaluation. Each sample was initially weighed and then immersed in phosphate-buffered saline (PBS) at 37 °C to replicate physiological conditions. Following an immersion period of 12 h, the films were carefully removed, dried at 100 °C, and reweighed. This process was repeated at the intervals of 24, 36, 48, and 72 h. The percentage weight loss of the films was calculated to quantify their degradation behavior, offering valuable insights into the stability and breakdown dynamics of the biopolymer matrices under simulated physiological conditions.

## 4. Conclusions

In this study, biopolymer films were prepared using polysaccharides, i.e., starch, chitosan, alginate, and their blends—starch with chitosan or alginate—for food packaging applications. As antimicrobial agents, various plant extracts were tested. Chestnut extract, showing the strongest antimicrobial activity against *S. epidermidis*, *E. coli*, and *C. albicans*, was chosen as the films’ antimicrobial agent. The films’ antimicrobial activity against *S. epidermidis*, *E. coli*, and *C. albicans* was evaluated, focusing on the effects of polymer matrix, plasticizers, cross-linking agents, and nanofiber cellulose. The chitosan-based films were more effective against *S. epidermidis* but ineffective against *C. albicans*. Adding nanofiber cellulose improved the mechanical and antimicrobial properties of chitosan films. The alginate films demonstrated the highest antimicrobial activity, especially against *E. coli*, with effectiveness enhanced by plasticizers like EPGOS and PGOS, which improved chestnut extract release as well as physical cross-linking, while the starch films with chemical cross-linking formed a compact structure that restricted agent diffusion, reducing antimicrobial efficacy. Blended films, i.e., STCH and STALG, showed improved activity against *S. epidermidis* compared to ST. The starch films displayed stronger antifungal activity against *C. albicans* than ALG. Balancing hydrophilicity and mechanical strength is also key to optimizing antimicrobial performance. These findings underscore the potential of these biodegradable films with antimicrobial properties as sustainable options for modern food preservation applications.

## Figures and Tables

**Figure 1 ijms-25-12580-f001:**
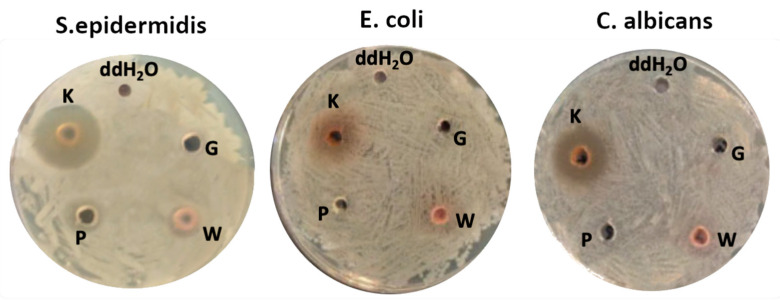
Antimicrobial activity of four commercially available extracts—chestnut (K), graviola (G), grape (W), and nettle (P)—tested against model microorganisms, including Gram-positive bacteria (*S. epidermidis*), Gram-negative bacteria (*E. coli*), and yeast (*C. albicans*). The control well contained deionized water (ddH_2_O).

**Figure 2 ijms-25-12580-f002:**
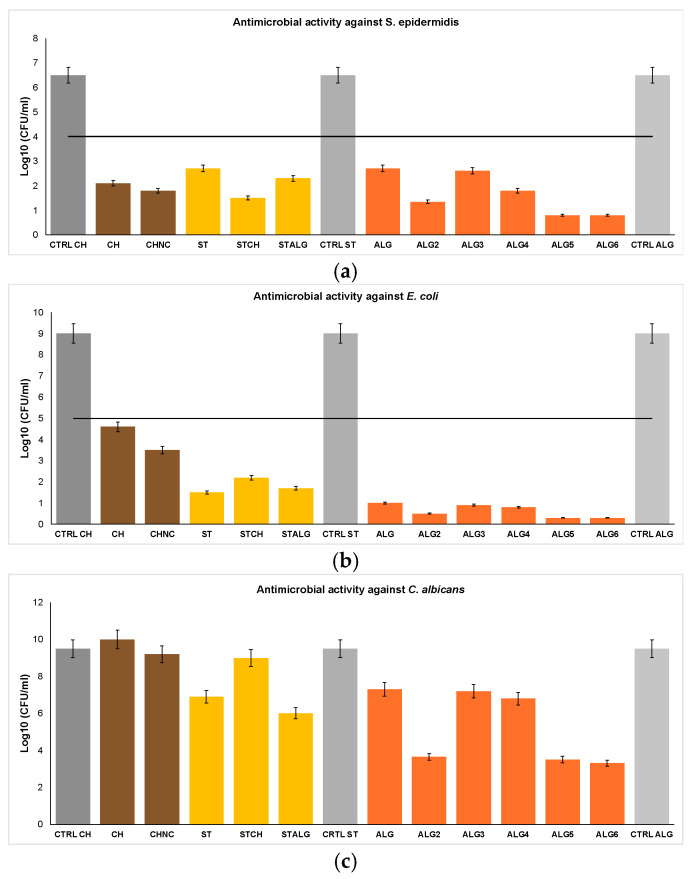
The effect of modified polysaccharide films on the viability of *S. epidermidis* cells (**a**), *E. coli* cells (**b**), and *C. albicans* (**c**). The threshold for antibacterial activity is represented by the horizontal line (**a**,**b**). Samples with values below this threshold indicate effective antibacterial action, while samples with values above this line demonstrate insufficient antibacterial efficacy. The list of abbreviations for the investigated films: CTRL CH: control sample for chitosan; CH: chitosan; CHNC: chitosan reinforced with nanofiber cellulose; ST: starch; STCH: a blend of starch and chitosan; STALG: a blend of starch and alginate; CTRL ST: control sample for starch films; ALG: alginate with glycerol; ALG2: alginate with epoxidized soybean oil; ALG3: alginate films with epoxidized palm oil; ALG4: alginate with mixed esters of propylene glycol and acetic acid; ALG5: alginate with mixed esters of propylene glycol, oleic acid, and succinic acid; ALG6: alginate with epoxidized mixed esters of propylene glycol, oleic acid, and succinic acid; CTRL ALG: control sample for alginate.

**Figure 3 ijms-25-12580-f003:**
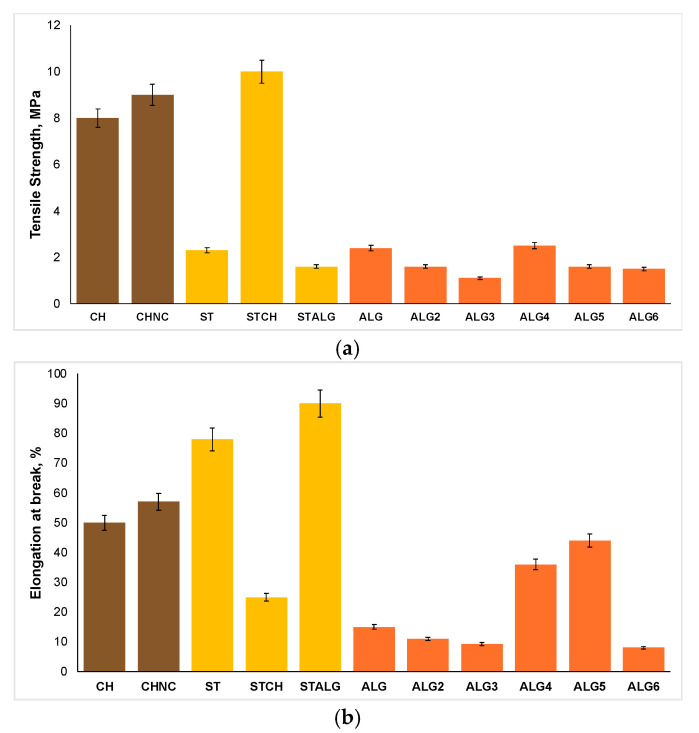
Tensile strength (**a**) and elongation at break (**b**) of polysaccharide films based on alginate (ALG), starch (ST), and chitosan (CH).

**Figure 4 ijms-25-12580-f004:**
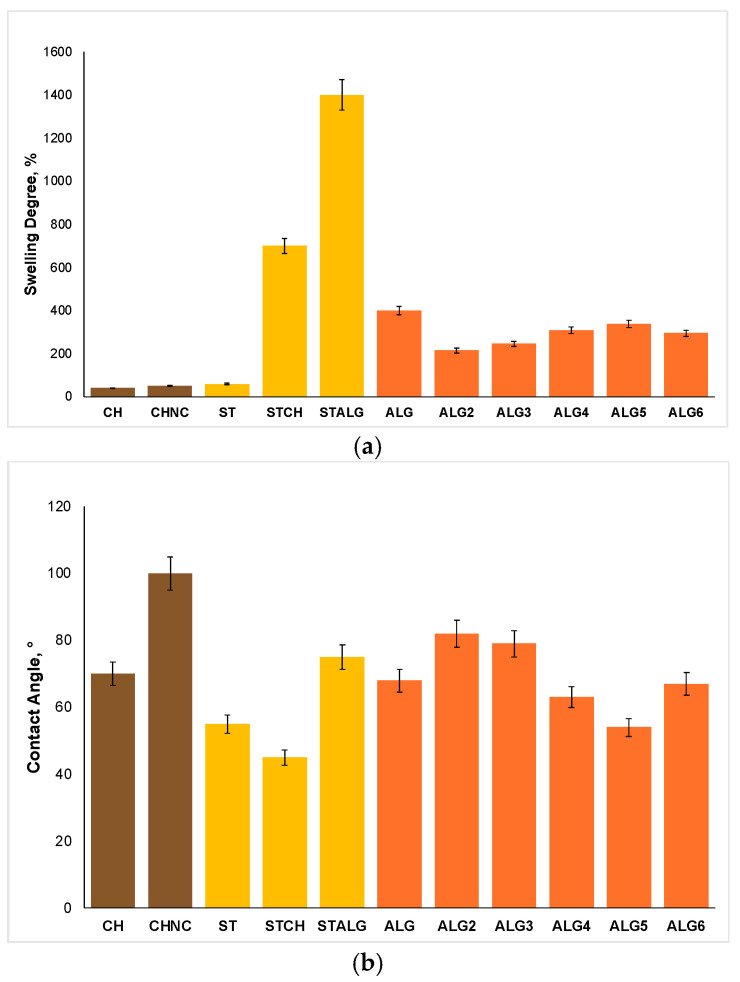
Swelling degree (**a**) and contact angle (**b**) of polysaccharide films based on alginate (ALG), starch (ST), and chitosan (CH).

**Figure 5 ijms-25-12580-f005:**
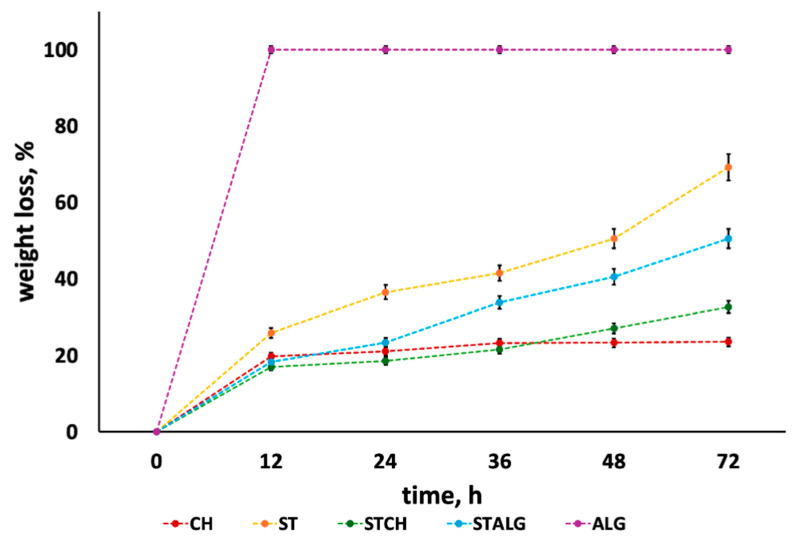
In vitro hydrodegradation of polysaccharide films based on alginate (ALG), starch (ST), and chitosan (CH).

**Figure 6 ijms-25-12580-f006:**
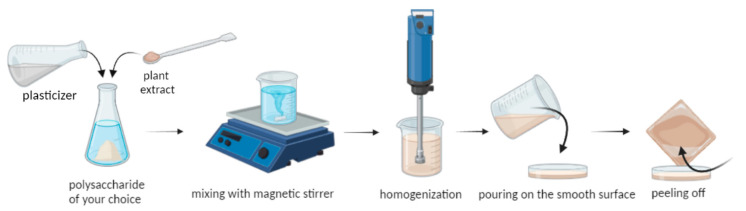
Scheme of the preparation of the antimicrobial films.

**Figure 7 ijms-25-12580-f007:**
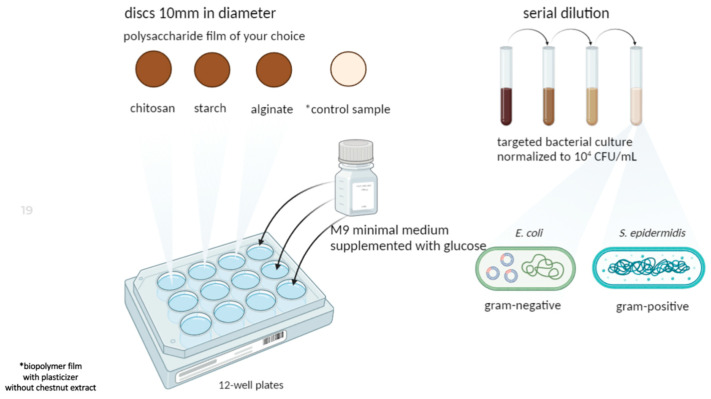
The scheme of antimicrobial activity measurement.

**Table 1 ijms-25-12580-t001:** Abbreviations for the investigated films and their blends, including added chestnut extract and plasticizers, and control samples (without the chestnut extract).

Chitosan films									
CH Chitosan with glycerol			CHNC Chitosan reinforced with nanofiber cellulose				CTRL CH Chitosan control without chestnut extract		
**Starch films**									
**ST** **Starch with glycerol**					**CTRL ST** **Starch control without chestnut extract**				
Alginate films									
ALG Alginate with glycerol	ALG2 Alginate with epoxidized soybean oil	ALG3 Alginate with epoxidized palm oil		ALG4 Alginate with mixed esters of propylene glycol and acetic acid		ALG5 Alginate with mixed esters of propylene glycol, oleic acid, and succinic acid		ALG6 Alginate with epoxidized mixed esters of propylene glycol, oleic acid, and succinic acid	CTRL ALG Alginate control without chestnut extract
Blends									
STCH Starch and chitosan with glycerol					STALG Starch and alginate with glycerol				

**Table 2 ijms-25-12580-t002:** Abbreviations of alginate films with different plasticizers.

Abbreviation	ALG	ALG2	ALG3	ALG4	ALG5	ALG6
Plasticizer	Commercial	Commercial	Commercial	Synthesized	Synthesized	Synthesized
Description	Glycerol	Epoxidized soybean oil	Epoxidized palm oil	Mixed esters of propylene glycol and acetic acid	Mixed esters of propylene glycol, oleic acid, and succinic acid	Epoxidized mixed esters of propylene glycol, oleic acid, and succinic acid

**Table 3 ijms-25-12580-t003:** Abbreviations of investigated films and their blends.

Abbreviations	ALG	ST	CH	CHNC	STCH	STALG
Sample	Sodium alginate films	Starch films	Chitosan films	Chitosan films reinforced with nanofiber cellulose	Starch and chitosan films	Starch and alginate films

## Data Availability

The data presented in this study are available on request from the corresponding author.

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
