# Peer review of "Improving Antimicrobial Properties of Biopolymer-Based Films in Food Packaging: Key Factors and Their Impact"

_ijms, 2024, doi:10.3390/ijms252312580_

Round 1

Reviewer 1 Report

Comments and Suggestions for Authors

This study present significant findings; however, some concerns should be considered as follows:

1. Title: The authors should mention for potential application in food packaging.

2. abstract: significant findings should be presented. The full name of bacterial strains should be stated when mentioning for the first time.

3. Intro: the authors should discuss the alarming escalations of pathogenic bacteria and their deleterious effects during food packaging. Why did you use all these biopolymers? The novelty should be justified.

4. Results: please correct as results and discussion. The aim and novelty should be stated at the beginning. Fig. 1: the authors should add Petri-dishes for their results of clear zones since all authors recognize the inhibition zone of bacteria. All figures should be improved in the PDF; however, they are fine in the compressed file. Line 204: the second proposed mechanism for chitosan was repeated in line 212, please revise. The biodegradation of the membrane should be also discussed since it affects the release of antimicrobial agents from the membrane.

5. Methods:  For plasticizers: the authors should study the morphology of the films by a scanning electron microscope, roughness, and colour intensity. The biodegradation of the membrane should be also examined. How did you select the amount of plasticizers and antimicrobial extract for biopolymer? The authors should add a section for statistical analysis, describing the number of replicates, test, software used in this study.

6. Conclusion: it too long, please summarize.  

Author Response

Reviewer #1: Thank you very much for your critical comments and thoughtful suggestions on our manuscript entitled “Improving Antimicrobial Properties of Biopolymer-Based Films: Key Factors and Their Impact”. We have studied the comments carefully and made corrections, which we hope will meet your approval. The changes have been highlighted in the manuscript using red. Below, we will provide our point-by-point explanations to the reviewers’ comments and questions:

  1. Title: The authors should mention for potential application in food packaging.

Answer:

We thank the Reviewer for the right comment. In response to the reviewer's suggestion, we have incorporated the potential application in food packaging into the article’s title: “Improving Antimicrobial Properties of Biopolymer-Based Films in Food Packaging: Key Factors and Their Impact.”

  1. abstract: significant findings should be presented. The full name of bacterial strains should be stated when mentioning for the first time.

Answer:

We thank the Reviewer for the right comment. As suggested by the Reviewer, we have highlighted the significant findings (lines 28-31) in the abstract and provided the full names of the bacterial strains (line 18).

  1. Intro: the authors should discuss the alarming escalations of pathogenic bacteria and their deleterious effects during food packaging. Why did you use all these biopolymers? The novelty should be justified.

Answer:

We thank the Reviewer for the right comment. As suggested by the Reviewer, we briefly discussed the alarming escalations of pathogenic bacteria and their deleterious effects during food packaging:

“Furthermore, the rise in pathogenic bacteria in food packaging has heightened concerns over food safety, as contamination during packaging, transport, and storage can lead to foodborne illnesses, spoilage, and economic losses. Pathogens like E. coli, S. aureus, Salmonella, L. monocytogenes, and C. albicans are often linked to outbreaks due to inadequate packaging that fails to prevent microbial growth [2,3]. In response, biopolymer-based films with natural antimicrobial agents offer a promising solution by actively inhibiting microbial contamination, thereby enhancing food safety and shelf life. Traditional packaging materials, while providing physical barriers, lack antimicrobial properties and may contribute to contamination by degrading into microplastics that support bacterial biofilm formation [4].”

We also have provided more explanation at the end of the Introduction, why all these specific biopolymers were chosen:

“All these biopolymers were selected to investigate the impact of various biopolymer matrices on antimicrobial properties. Originating from diverse sources, they are widely studied in the literature, allowing us to benchmark our biopolymer-based films against existing research.

To characterize antimicrobial activity properly, we also take into account other factors that could affect the performance of investigated films. According to previous studies [15–19], achieving antimicrobial effectiveness in films requires adding specific modifiers into the film matrix, such as nanofillers, cross-linking agents, plasticizers as well as plant extracts. These modifiers are widely available and well-described in the literature for their effectiveness in enhancing antimicrobial activity. We selected these additives to compare published findings with our experimental results. Films’ antimicrobial properties were examined against the most common Gram-positive and Gram-negative bacteria, as well as yeast. The specific impact of each additive is discussed. Furthermore, mechanical and hydrophilic properties were determined in the case of antimicrobial activity.”

  1. Results: please correct as results and discussion. The aim and novelty should be stated at the beginning.

Fig. 1: the authors should add Petri-dishes for their results of clear zones since all authors recognize the inhibition zone of bacteria. All figures should be improved in the PDF; however, they are fine in the compressed file. Line 204: the second proposed mechanism for chitosan was repeated in line 212, please revise.

The biodegradation of the membrane should be also discussed since it affects the release of antimicrobial agents from the membrane.

Answer:

We thank the Reviewer for the right comment. As suggested by the Reviewer, we have highlighted the aim and novelty of our study in the beginning of Results section:

“The aim of this study was to develop and optimize antimicrobial biopolymer-based films using polysaccharides—starch, chitosan, alginate, and their blends—for sustainable food packaging applications. The novelty of this research lies in its systematic evaluation of various modifiers, including plant extracts, plasticizers, cross-linking agents, and nanofillers as well as their combined impact on enhancing the antimicrobial, mechanical, and hydrophilic properties of these films. Findings from this study demonstrate that these modified biopolymer-based films exhibit promising antimicrobial efficacy, positioning them as viable candidates for future eco-friendly food packaging solutions.”

As suggested by the Reviewer, we added real-life photo and its description as Figure 1.

As suggested by the Reviewer, we described and discussed the films’ biodegradation in the Methods and Results section:

“4.8 Measurement of Biodegradation

The biodegradability of the films was tested using a soil degradation method. Small dry sample pieces (2x3 cm) were buried 5 cm below the soil surface in plastic containers. To maintain soil moisture at 40%, about 10 ml of water was added to the containers each day. After incubating for 15 days at 25 ± 1.0 °C, the samples' final weight was measured. The rate of degradation in the soil was determined by calculating the percentage of mass lost compared to the initial weight of the samples.

2.4 Biodegradation of biopolymer-based films

The study on soil degradation of the prepared films revealed that all films fully de-composed within 15 days. Although we used three types of biopolymer matrix, the biodegradability was similar. This is likely because all the matrices examined are polysaccharides, giving them similar characteristics. Moreover, the investigated modifiers come from natural sources, thus they didn’t worsen the degradation rate.”

  1. Methods:  For plasticizers: the authors should study the morphology of the films by a scanning electron microscope, roughness, and colour intensity. The biodegradation of the membrane should be also examined. How did you select the amount of plasticizers and antimicrobial extract for biopolymer? The authors should add a section for statistical analysis, describing the number of replicates, test, software used in this study.

Answer:

We thank the Reviewer for the right comment.

In our previous research: https://doi.org/10.1038/s41598-023-38794-3 , we examined the film morphology using scanning electron microscopy, as well as assessed film roughness and colour intensity. This research did not demonstrate any effect on the antimicrobial properties of the films; thus we decided not to include it in the current study.

Nevertheless, we conducted the One-Way ANOVA test and two-tailed Student's t-test (JMP software v.5; SAS Institute Inc., Cary, NC, U.S.A.) to compare the samples against model organisms i.e., S. epidermidis, E. coli, and C. albicans. The resulting p-values were all smaller than 0.05, indicating statistically significant differences between the tested samples such as chitosan, starch, and alginate. These p-values could imply that S. epidermidis, E. coli, or C. albicans have different responses to the added chestnut extract.

  1. Conclusion: it too long, please summarize.  

Answer:

We thank the Reviewer for the right comment.

According to the Reviewer’s suggestion, we shortened conclusions in some places. However, because of the suggestions of another Reviewers, we expanded conclusions with information about our findings, highlighting the study’s limitations, and addressing the need for further validation. Additionally, the Reviewer emphasized the importance of contextualizing potential practical limitations of implementing these biopolymer films in food packaging, such as challenges with scalability and the stability of antimicrobial properties under varying environmental conditions.

Reviewer 2 Report

Comments and Suggestions for Authors

Humanity has achieved significant health advancements through the progress of medical and pharmaceutical technology. However, the responsibility for these advancements also lies with us. This paper not only introduces a valuable antimicrobial peptide but also emphasizes its biodegradable characteristics using environmentally friendly materials. This study holds high commercial and pharmaceutical value. I would like to offer a few additional suggestions to further enhance this paper.

1. Please describe the advantages of using environmentally friendly natural polymers compared to synthetic polymers, such as plastics.

2. Is it possible to explain the benefits of using gum-based materials over starch-based ones?

3. Could you add a real-life example photo to Figure 1?

4. The research narrative driven by the method is excellent overall but may benefit from being more concise, particularly in sections 2.2.1 through 2.2.4.

5. The relationship between the contact angle and the effect of antimicrobial agents is not sufficiently highlighted in the results section.

Author Response

Reviewer #2: Thank you very much for your critical comments and thoughtful suggestions on our manuscript entitled “Improving Antimicrobial Properties of Biopolymer-Based Films: Key Factors and Their Impact”. We have studied the comments carefully and made corrections, which we hope will meet your approval. The changes have been highlighted in the manuscript using red. Below, we will provide our point-by-point explanations to the reviewers’ comments and questions:

  1. Please describe the advantages of using environmentally friendly natural polymers compared to synthetic polymers, such as plastics.

Answer:

We thank the Reviewer for the right comment. As suggested by the Reviewer, we have provided the description of advantages of using environmentally friendly natural polymers in Introduction Section:

“Derived from renewable resources such as plant biomass and animal waste products, natural polymers offer a sustainable alternative to fossil-based plastics. Natural polymers are inherently biodegradable, breaking down under atmospheric factors or microbial action into low-molecular-weight compounds that cause no harm to the environment. In contrast, synthetic polymers like plastics degrade into microplastics, which may accumulate in ecosystems and enter food chains, posing potential risks to wildlife and human health. These qualities position natural polymers as a preferred choice for developing safe, eco-friendly antimicrobial films for food packaging materials.”

  1. Is it possible to explain the benefits of using gum-based materials over starch-based ones?

Answer

According to the Reviewer’s suggestion, we explain the benefits of using gum-based materials over starch-based ones in the text below:

We agree with the Reviewer that gum-based materials offer several advantages over starch-based options for food packaging. They provide greater mechanical strength and flexibility, even under temperature changes, unlike starch, which can gelatinize and lose effectiveness when heated. Gums achieve higher viscosity at lower concentrations, making them efficient thickeners and stabilizers. They also provide better barrier properties, forming excellent moisture and grease barriers. Despite these gum’s advantages, it is not biodegradable like starch.

On the other side, starch materials have worse mechanical and water interaction properties. However, disadvantage of starch film can be improved by creating blends or/and incorporating various modifiers i.e., nanoparticles, cross-linking agents, plasticizers as well as plant extracts. Such modifications enhance not only mechanical and water interaction properties, but also antimicrobial ones. Considering these benefits, in our study we added to starch films chestnut extract, and oxidized sucrose. We also prepared blends with chitosan and alginate. Our research proves that such attitude towards starch is beneficial for future application of such material in food packaging industry.

  1. Could you add a real-life example photo to Figure 1?.

Answer:

We thank the Reviewer for the right comment. As suggested by the Reviewer, we added real-life photo and its description as Figure 1.

  1. The research narrative driven by the method is excellent overall but may benefit from being more concise, particularly in sections 2.2.1 through 2.2.4.

Answer:

We thank the Reviewer for the right comment. As suggested by the Reviewer, we have made revisions to enhance conciseness in the Sections from 2.2.1 to 2.2.4.

  1. The relationship between the contact angle and the effect of antimicrobial agents is not sufficiently highlighted in the results section.

Answer:

We thank the Reviewer for the right comment. As suggested by the Reviewer, we have improved this section to emphasize the relationship more clearly between contact angle and the effect of antimicrobial agents:

“The water contact angle (Figure 4b) plays a key role in influencing the antimicrobial properties of films by determining their hydrophilicity or hydrophobicity, which affects the release of antimicrobial agents, consequently enhancing the films’ antimicrobial effectiveness. A contact angle (θ) below 90° denotes a hydrophilic surface, while angle value above 90° indicates hydrophobicity.”

Reviewer 3 Report

Comments and Suggestions for Authors

This manuscript explores enhancing the antimicrobial properties of biopolymer-based films by incorporating various modifiers, including plant extracts, plasticizers, and nanofillers, targeting applications in food packaging. Although the topic is timely and relevant, the article has several areas where improvements are essential to achieve the standards required for publication. My suggestions are listed below:

In the introduction, provide more background on why these specific modifiers were chosen based on their properties and how they are hypothesized to enhance the films’ antimicrobial effects.

Although the study aims to enhance biopolymer-based films, it lacks a clear hypothesis regarding which modifiers are expected to have the most significant impact or why certain materials were chosen. Adding a hypothesis would strengthen the scientific rationale.

Figure 2 (lines 169-172) attempts to show antimicrobial activity against various organisms, but the text fails to discuss these results systematically, leading to confusion. A more structured narrative connecting each data set with its interpretation is necessary.

The abbreviations for film samples (e.g., ALG5, CHNC) are inconsistently introduced, as seen in lines 176-180, making it challenging to track the composition of each sample across sections. Providing a concise table of abbreviations at the beginning of the results section would improve readability.

While some mechanisms of antimicrobial action are briefly mentioned (lines 195-209), the manuscript lacks a deep, cohesive discussion on how specific components (e.g., chestnut extract, calcium chloride) contribute to antimicrobial effects. For example, the role of tannins from chestnut extract in disrupting bacterial cell membranes is only superficially addressed, and details about their interaction with bacterial structures are missing.

The differences in antimicrobial efficacy between plant extracts, such as chestnut versus nettle, are mentioned (lines 136-151) but not fully explored. A comparative discussion analyzing why certain extracts perform better would enhance the manuscript's scientific rigor.

Although p-values are presented, there is minimal discussion regarding the statistical methodology used to ensure reliability, especially in sections comparing antimicrobial performance across treatments (lines 153-163).

The discussion could better contextualize the potential practical limitations of implementing such biopolymer films in food packaging, such as scalability issues or the stability of antimicrobial properties under varying environmental conditions.

The conclusion (lines 551-559) generalizes the study’s findings by implying broad applicability of these films for food preservation. This claim is premature given the lack of real-world testing. Instead, the conclusion should focus on highlighting the study's limitations and the need for further validation.

Author Response

Reviewer #3: Thank you very much for your critical comments and thoughtful suggestions on our manuscript entitled “Improving Antimicrobial Properties of Biopolymer-Based Films: Key Factors and Their Impact”. We have studied the comments carefully and made corrections, which we hope will meet your approval. The changes have been highlighted in the manuscript in red. Below, we will provide our point-by-point explanations to the reviewers’ comments and questions:

  1. In the introduction, provide more background on why these specific modifiers were chosen based on their properties and how they are hypothesized to enhance the films’ antimicrobial effects.

Answer:

We thank the Reviewer for the right comment. As suggested by the Reviewer, we have provided more background at the end of the Introduction, why these specific modifiers were chosen:

“According to previous studies [15–19], achieving antimicrobial effectiveness in films requires adding specific modifiers into the film matrix, such as nanofillers, cross-linking agents, plasticizers as well as plant extracts. These modifiers are widely available and well-described in the literature for their effectiveness in enhancing antimicrobial activity. We selected these additives to compare published findings with our experimental results. Films’ antimicrobial properties were examined against the most common Gram-positive and Gram-negative bacteria, as well as yeast. The specific impact of each additive is discussed. Furthermore, mechanical and hydrophilic properties were determined in the case of antimicrobial activity.”

  1. Although the study aims to enhance biopolymer-based films, it lacks a clear hypothesis regarding which modifiers are expected to have the most significant impact or why certain materials were chosen. Adding a hypothesis would strengthen the scientific rationale.

Answer

We thank the Reviewer for the right comment. As suggested by the Reviewer, we have described at the end of the Introduction that certain materials were chosen, because they are widely available and well-described in the literature for their effectiveness in enhancing antimicrobial activity. Thus, we selected these additives to compare published findings with our experimental results.

  1. Figure 2 (lines 169-172) attempts to show antimicrobial activity against various organisms, but the text fails to discuss these results systematically, leading to confusion. A more structured narrative connecting each data set with its interpretation is necessary.

Answer:

We thank the Reviewer for the right comment. As suggested by the Reviewer, we have added another description about Figure 2, explaining antimicrobial properties of the investigated films:

“The antimicrobial properties of the investigated films, summarized in Figure 2, are shown against representative model microorganisms, specifically targeting Gram-positive bacteria (S. epidermidis) in Figure 2a, Gram-negative bacteria (E. coli) in Figure 2b, as well as yeast (C. albicans) in Figure 2c.”

  1. The abbreviations for film samples (e.g., ALG5, CHNC) are inconsistently introduced, as seen in lines 176-180, making it challenging to track the composition of each sample across sections. Providing a concise table of abbreviations at the beginning of the results section would improve readability.

Answer:

We thank the Reviewer for the right comment. As suggested by the Reviewer, we created additional table of abbreviations at the beginning of the Results section.

  1. While some mechanisms of antimicrobial action are briefly mentioned (lines 195-209), the manuscript lacks a deep, cohesive discussion on how specific components (e.g., chestnut extract, calcium chloride) contribute to antimicrobial effects. For example, the role of tannins from chestnut extract in disrupting bacterial cell membranes is only superficially addressed, and details about their interaction with bacterial structures are missing.

Answer:

We thank the Reviewer for the right comment. As suggested by the Reviewer, in Section 2.2.1 we delved into antimicrobial effects of specific components such as nanofiber cellulose, chestnut extract, calcium chloride, oxidized sucrose as well as plasticizers.

  1. The differences in antimicrobial efficacy between plant extracts, such as chestnut versus nettle, are mentioned (lines 136-151) but not fully explored. A comparative discussion analysing why certain extracts perform better would enhance the manuscript's scientific rigor.

Answer:

We thank the Reviewer for the right comment. As suggested by the Reviewer, we have compared the efficacy of chestnut extract with that of other extracts and provided an explanation for the observed phenomenon.

“For instance, chestnut extract has stronger antimicrobial efficacy than nettle [23], grape [18] and graviola [24] extract owing to its high tannin content, particularly ellagitannins [25], which effectively disrupt microbial cells by binding to proteins, interfering with enzymes, and damaging cell membranes [26]. These tannins also deprive bacteria of essential nutrients, providing broad-spectrum activity against both Gram-positive and Gram-negative bacteria and reducing the risk of resistance.”

  1. Although p-values are presented, there is minimal discussion regarding the statistical methodology used to ensure reliability, especially in sections comparing antimicrobial performance across treatments (lines 153-163).

Answer:

We thank the Reviewer for the right comment. As suggested by the Reviewer, we added statistical test, One-Way ANOVA to ensure reliability of films’ antimicrobial performance.

“Additionally, the One-Way ANOVA test was conducted to compare the samples against model organisms i.e., S. epidermidis, E. coli, and C. albicans. The resulting p-values were all smaller than 0.05, indicating statistically significant differences between the tested samples such as chitosan, starch, and alginate. These p-values could imply that S. epidermidis, E. coli, or C. albicans have different responses to the extract”.

  1. The discussion could better contextualize the potential practical limitations of implementing such biopolymer films in food packaging, such as scalability issues or the stability of antimicrobial properties under varying environmental conditions.

Answer:

We thank the Reviewer for the right comment. As suggested by the Reviewer, we added appropriate information to the Conclusions section:

“However, real-world application and testing are needed to fully understand some potential limitations related to films’ weaker structural integrity, varying degrees of swelling degree, and diffusion rates of antimicrobial agents. This practical evaluation will clarify whether biopolymer films can consistently meet the demands of large-scale food packaging. If these antimicrobial properties can be enhanced, such films could play a significant role in the development of biodegradable packaging for the food industry, helping to prevent oxidation and spoilage and ultimately extending the shelf life of food products.”

  1. The conclusion (lines 551-559) generalizes the study’s findings by implying broad applicability of these films for food preservation. This claim is premature given the lack of real-world testing. Instead, the conclusion should focus on highlighting the study's limitations and the need for further validation.

Answer:

We thank the Reviewer for the right comment. As suggested by the Reviewer, we focused more on our results in Conclusions Section and highlighted the study’s limitations with the need for further validation:

“Nevertheless, chitosan films were ineffective against C. albicans due to limited chestnut extract release in neutral pH. Chitosan as a weak base is insoluble in water and organic solvents at neutral pH, limiting its intrinsic antifungal activity and slowing the release of chestnut extract in a biopolymer matrix.”

“Moreover, the alginate’s matrix has hydrophilic properties, which promotes the migration of additives like chestnut extract to the film’s surface. When swelled in water, alginate creates space between polymer chains, facilitating chestnut extract movement. Once on the surface, the extract becomes active, inhibiting bacterial growth, with tannins playing a key role by prolonging the bacterial lag phase and slowing growth.”

“The interaction between cellulose and chitosan enhances nanofiber dispersion, reinforcing the film’s structure and reducing brittleness. Chitosan nanofillers also enhance antimicrobial properties by reducing bacterial motility and preventing biofilm formation. Additionally, nanofiber cellulose increases film stability and moisture resistance, making it suitable for advanced applications.”

“However, real-world application and testing are needed to fully understand some potential limitations related to films’ weaker structural integrity, varying degrees of swelling degree, and diffusion rates of antimicrobial agents. This practical evaluation will clarify whether biopolymer films can consistently meet the demands of large-scale food packaging. If these antimicrobial properties can be enhanced, such films could play a significant role in the development of biodegradable packaging for the food industry, helping to prevent oxidation and spoilage and ultimately extending the shelf life of food products.”

Round 2

Reviewer 1 Report

Comments and Suggestions for Authors

The authors exerted their efforts for addressing the previous comments; however, some comments should still be considered as follows:

1. Abstract: significant findings should be presented.

2. Line 138: does Fig. 1 refer to a scheme? Please correct. Line 140: Supplementary information, the authors did not upload SI to review. I presume the authors did not correct it form the previous version. Legend of Fig. 1: please revise and correct what each well refers to.

3. Lines 130, 519, and 520: the name of bacterial strains should be italic. Please revise these mistakes throughout the manuscript.

4. Section 2.4. the degradation of biopolymer carriers for extracts or drugs should be conducted by immersing them in PBS at 37 °C for predetermined time intervals and then weigh the loss of their weights. Please check this article for performing the test (https://doi.org/10.3390/pharmaceutics14122649).

5. Conclusion: it is too long and the authors did not consider my previous comment. It should be 120-200 words.  

Author Response

Reviewer #1: Thank you very much for your critical comments and thoughtful suggestions on our manuscript entitled “Improving Antimicrobial Properties of Biopolymer-Based Films: Key Factors and Their Impact”. We have studied the comments carefully and made corrections, which we hope will meet your approval. The changes have been highlighted in the manuscript using red. Below, we will provide our point-by-point explanations to the reviewers’ comments and questions:

  1. Abstract: significant findings should be presented.

Answer:

We thank the Reviewer for the right comment. As suggested by the Reviewer, we have presented significant findings in the end of Abstract section:

“All these findings highlight the potential of these biodegradable films for food packaging, offering enhanced antimicrobial activity that prolongs shelf life and reduces spoilage, making them promising candidates for sustainable food preservation.”

  1. Line 138: does Fig. 1 refer to a scheme? Please correct. Line 140: Supplementary information, the authors did not upload SI to review. I presume the authors did not correct it form the previous version. Legend of Fig. 1: please revise and correct what each well refers to.

Answer:

We thank the Reviewer for the right comment. In accordance with the Reviewer’s suggestion, we have revised the relevant section of the text, which refers to the representative photograph from the Inhibition Zone Assay results:

Figure 1. shows the representative photograph displaying the Inhibition Zone Assay results against S. epidermidis, E. coli and C. albicans.”

  1. Lines 130, 519, and 520: the name of bacterial strains should be italic. Please revise these mistakes throughout the manuscript.

Answer:

We thank the Reviewer for the right comment. As suggested by the Reviewer, we have revised these mistakes throughout the manuscript.

  1. Section 2.4. the degradation of biopolymer carriers for extracts or drugs should be conducted by immersing them in PBS at 37 °C for predetermined time intervals and then weigh the loss of their weights. Please check this article for performing the test (https://doi.org/10.3390/pharmaceutics14122649).

Answer:

We thank the Reviewer for the right comment. As suggested by the Reviewer, we have conducted the degradation study by our films in PBS at 37 °C for 72 hours. We added the following text:

In Materials and Methods section:

“We also conducted the hydrolytic degradation of investigated films, samples with a surface area of 1 cm²—composed of chitosan, starch, alginate, as well as their blends – starch with chitosan or alginate—underwent a systematic evaluation. Each sample was initially weighed and then immersed in phosphate-buffered saline (PBS) at 37 °C to replicate physiological con-ditions. Following an immersion period of 12 hours, the films were carefully removed, dried at 100 °C, and reweighed. This process was repeated at intervals of 24, 36, 48, and 72 hours. The percentage weight loss of the films was calculated to quantify their degradation behavior, of-fering valuable insights into the stability and breakdown dynamics of the biopolymer matrices under simulated physiological conditions.”

In Results section:

“The degradation of the investigated films was also evaluated by immersing them in PBS at 37 °C. The results, presented in Figure 7, demonstrate how the type of biopolymer matrix influences the degradation behavior of these films. The film with the highest biodegradability was alginate, which decomposed in less than 12 hours. Other films did not degrade completely within 72 hours, exhibiting different weight loss rates. The most stable film was chitosan, with a weight loss of 23.54 ± 1.54% after 72 hours. In contrast, the starch film exhibited a weight loss three times greater than chitosan, amounting to 69.17 ± 1.83%. Blending matrices combines the properties of each biopolymer. For the starch-chitosan blend, the weight loss was intermediate between the individual polymers, measuring 32.76 ± 1.65%. A similar trend was observed for the starch-alginate blend, which exhibited a weight loss of 50.54 ± 1.72%. Comparable results for biopolymer degradation have been previously reported in [78], [79], [80], [81].”

  1. Conclusion: it is too long and the authors did not consider my previous comment. It should be 120-200 words.

We thank the Reviewer for the right comment. In accordance with the Reviewer’s suggestion, we have significantly shortened the Conclusions section:

Answer:

“In this study, biopolymer films were prepared using polysaccharides, i.e., starch, chitosan, alginate, and their blends - starch with chitosan or alginate for food packaging applications. As antimicrobial agents, various plant extracts were tested. Chestnut extract, showing the strongest antimicrobial activity against S. epidermidis, E. coli and C. albicans, was chosen as the films’ antimicrobial agent. The films’ antimicrobial activity against S. epidermidis, E. coli, and C. albicans was evaluated, focusing on the effects of polymer matrix, plasticizers, cross-linking agents, and nanofiber cellulose. Chitosan-based films were more effective against S. epidermidis but ineffective against C. albicans. Adding nanofiber cellulose improved mechanical and antimicrobial properties of chitosan films. Alginate films demonstrated the highest antimicrobial activity, especially against E. coli, with effectiveness enhanced by plasticizers like EPGOS and PGOS, which improved chestnut extract release as well as physical cross-linking. While starch films with chemical cross-linking formed a compact structure that restricted agent diffusion, reducing antimicrobial efficacy. Blended films i.e., STCH and STALG showed improved activity against S. epidermidis compared to ST. Starch films displayed stronger antifungal activity against C. albicans than ALG. Balancing hydrophilicity and mechanical strength is also key to optimizing antimicrobial performance. These findings underscore the potential of these biodegradable films with antimicrobial properties as sustainable options for modern food preservation applications.”

Reviewer 3 Report

Comments and Suggestions for Authors

The authors took into account the recommendations of the reviewers, I recommend the publication of the article

Author Response

Thank you for your time and valuable feedback and for accepting our manuscript.

Round 3

Reviewer 1 Report

Comments and Suggestions for Authors

The authors addressed comments properly.